# REPANA: Reasoning Path Navigated Program Induction for Universally Reasoning over Heterogeneous Knowledge Bases

## Abstract

Program induction is a typical approach that helps Large Language Models (LLMs) in complex knowledge-intensive question answering over knowledge bases (KBs) to alleviate the hallucination of LLMs. However, the accurate program induction usually requires a large number of high-quality parallel data of a specific KB, which is difficult to acquire for many low-resource KBs. Additionally, due to heterogeneity of questions and KB schemas, the transferability of a model trained on a single dataset is poor. To this end, we propose **REPANA**, a reasoning path navigated program induction framework that enables LLMs to reason over heterogeneous KBs. We decouple the program generation capability into perceiving the KB and mapping questions to program sketches. Accordingly, our framework consists of two main components. The first is an LLM-based navigator, which retrieves reasoning paths of the input question from the given KB. The second is a KB-agnostic parser trained on data from multiple heterogeneous datasets, taking the navigator's retrieved paths and the question as input and generating the corresponding program. Experiments show that REPANA exhibits strong generalization and transferability. It can directly perform inference on datasets not seen during training, outperforming other SoTA low-resource methods and even approaching the performance of supervised methods.

## 1 Introduction

Recently, incorporating knowledge bases (KBs) as external knowledge to augment large language models (LLMs) (Brown et al., 2020; OpenAI, 2023) in knowledge-intensive question answering has become a typical approach (Jiang et al., 2023a; Li et al., 2023b; Xie et al., 2022) to address the challenge of hallucination (Huang et al., 2023), namely the tendency that LLMs confidently make up factually incorrect answers.

In this light, recent work can roughly be categorized in to two types. The first is program induction (PI) method (Gu et al., 2021) that translate a given natural language question into an interpretable logical form, such as KoPL (Cao et al., 2022a) or SPARQL (Pérez et al., 2006), which is executable against the KB for getting the answer. Multiple techniques are utilize to boost the performance, such as retrieval augmentation (Ye et al., 2022), in-context learning (Li et al., 2023a), instruction tuning (Luo et al., 2023) and so on. However, to achieve a strong performance, these works typically require training on a single KB with a large amount of question-program pairs, which are difficult to obtain by manual annotation. The second is the agent-based method (Jiang et al., 2023a; Sun et al., 2024; Gu et al., 2023) that use LLMs to dynamically explore the knowledge graph step by step with predefined actions like extract relations and entities. In this way, the LLMs can help make decision at every reasoning step. However, these methods are restricted by the predefined action, not able to perform complex operations such as comparison and calculation. Although Jiang et al. (2024) defines a more comprehensive toolbox, the use of complex tool combinations is essentially equivalent to the program, and it still rely on large amounts of training data.

As shown in Figure 1, existing PI methods heavily rely on high-quality parallel data and lack transferability across heterogeneous datasets; meanwhile, agent-based methods can only handle limited types of complex questions, and requires at least one topic entity in the question. To tackle these

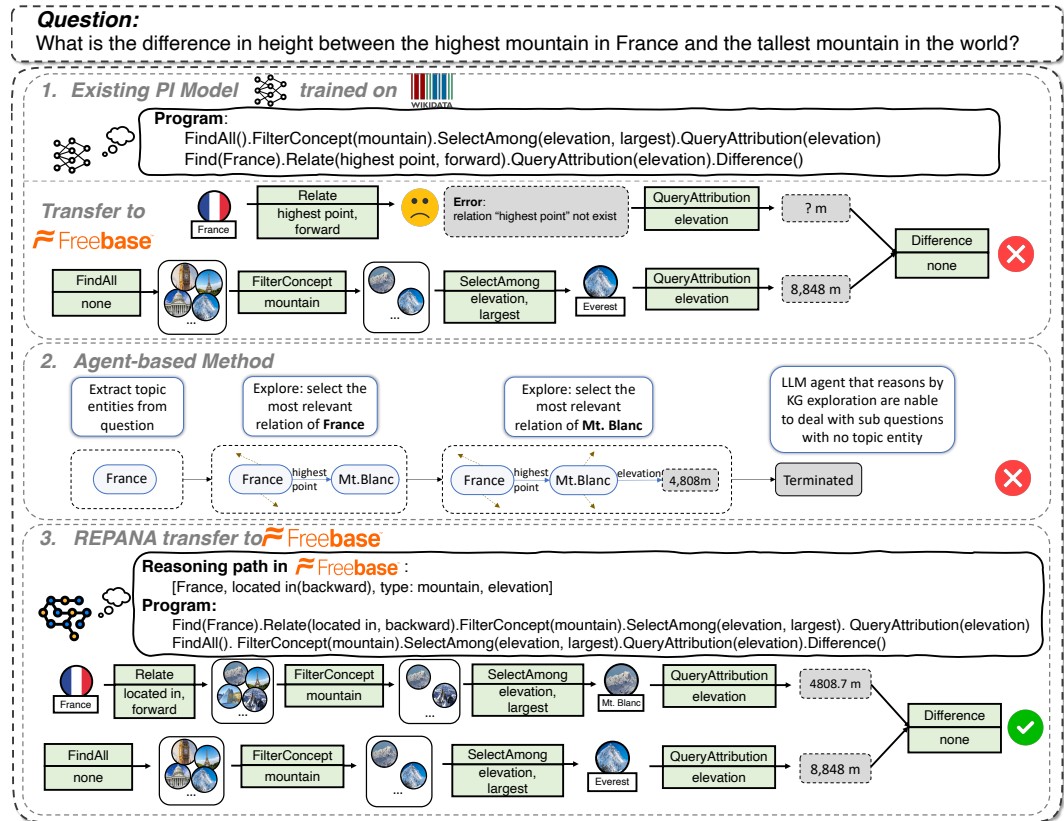

Figure 1: The PI model trained on datasets built on Wikidata fails to reason on Freebase since the label "highest point" in Freebase is not included in its schema. The agent-based method fails to deal with the sub-question that without a top entity to start exploration. REPANA avoids the shortcomings of both methods. It can generate the correct relation label in another KB because the parser is given the right schema items from the path. Although the lack of topic entity also affect the beam-search-like retrieval of reasoning path in the first stage, the trained parser of the second stage can partially address the issue since the parser knows the possibly correct program sketch.

problems, inspired by the idea indicated by recent studies (Cao et al., 2022b) that the ability to map questions to program sketches (namely the composition of program functions) is only depending on the structure of language and transferable across KBs, we propose to address the above challenges by training a KB-agnostic universal parsing model, along with a navigation module that retrieves the specific reasoning path information from KB. In this paper, we propose **REPANA**, the reasoning path navigated framework that enables LLMs to reason over heterogeneous questions and KBs.

Unlike existing PI models which generate program by simultaneously learning the schema of KB and the mapping from question to program from the parallel data, REPANA decouples and reconstructs the process into two parts: perceiving the schema of KB and mastering the mapping from questions to program sketches. To be specific, there are two key modules in the framework. The first is the LLM-based **KB navigator** that aims to locate and return the reasoning path that contains the necessary program arguments such as relation labels in KB, enabling the system to partly perceive the schema of the KB. The other is the **KB-agnostic parser** trained on rich-resource KB, primarily learning the program's syntax and grammar and mapping from question to program sketches, without deeply fitting a specific KB.

Through of this naval two-stage design, we ensure the retrieval efficiency and accuracy thus reduce the introduced noise, while enabling the model to perform reasoning on low-resource knowledge bases without the need for training. Specifically, in the first stage, we design an LLM-based KB-walk search strategy similar to beam search. Starting from the root entity of question, the navigator can accurately select the most relevant relations to the question in each walking step, and finally return

a most viable path through backtracking. In the second stage, we fist train the LLM parser on the datasets that are based on the rich-resource KB. The parser takes both the question and the retrieved reasoning path as input, selects the necessary elements from the reasoning path as arguments, and generate the final program. Since the LLM-based KB navigator does not require extra training and the KB-agnostic parser only need to be trained once, REPANA addresses the issue of transferability, thereby alleviating the shortage of annotated data.

In the experiment, we sample the training data from KQA Pro (Cao et al., 2022a), which is based Wikidata (Vrandecic & Krötzsch, 2014), as the rich-resource KB, then try to transfer to other datasets based on different KB, such as GrailQA (Gu et al., 2021) that based on Freebase (Bollacker et al., 2008). We first evaluate REPANA on KQA Pro. The results show that REPANA is comparable to the performance of several supervised SoTA methods with fewer training data. Then we evaluate REPANA on other unseen datasets during training (GrailQA, WebQSP, ComplexWebQuestions, MetaQA, etc.). The results demonstrate that REPANA outperforms SoTA low-resource PI methods with up to 20 times smaller backbone model. Our contributions in this paper include: (1) proposing REPANA, a novel reasoning path navigated program induction framework that enables LLMs to universally reason over the low-resource datasets; (2) demonstrating the effectiveness and indispensability of our decoupled two-stage generation strategy through extensive experiments and ablation studies.

## 2 RELATED WORK

### 2.1 KNOWLEDGE BASE QUESTION ANSWERING

Knowledge Base Question Answering (KBQA) aims to answer natural language questions based on fact triples stored in the KB, such as Wikidata Vrandecic & Krötzsch (2014) and Freebase Bollacker et al. (2008). Typical methods for solving KBQA problems can be broadly divided into two groups: (1) program induction based method, which converts questions into executable logical forms called program. The programs are usually generated by step-by-step graph searching Gu et al. (2021); Jiang et al. (2023b;a); Gu et al. (2023) or by sequence-to-sequence model trained with parallel data Ye et al. (2022); Cao et al. (2022b); Shu et al. (2022); Yu et al. (2023); Luo et al. (2023); (2) information retrieval based method, which usually output the answer by retrieving triples and subgraphs related to the question from KB or embedded memory Sun et al. (2019); Shi et al. (2021); Zhang et al. (2022); Oguz et al. (2022); Dong et al. (2023). Recent works Jiang et al. (2023a;b); Sun et al. (2024) that leverage LLMs as agents to explore the KB also belongs to this group. They search the KB by step-by-step prompting the LLMs for next action. However, they can only handle a limited range of questions with their limited pre-defined actions, and cannot easily adapt between different KBs.

### 2.2 LOW-RESOURCE PROGRAM INDUCTION

One line of work is utilizing the in-context-learning ability of LLMs to perform few-shot program generation Li et al. (2023a); Bogin et al. (2023); Gu et al. (2023), but their performance usually are limited by the context window. They also face challenges in distinguishing similar schema items in the KB, causing models to overly rely on post-processing steps like relation linking. A variation Li et al. (2024) is using LLMs to few-shot generate question given the program, then training a smaller model with the generated pseudo pairs. But their programs either comes from existed datasets or templates, leading to insufficient diversity and scalability.

The other line is program transfer method, which leverage the annotation from rich-resource KB to aid program induction for low-resource KB. Cao et al. proposed a two-stage parsing framework that first generate the program sketch, then fill in the rest arguments by searching the KB. However, due to the heterogeneity, it performs poorly without fine-tuning using annotated data from low-resource KB. Zhang et al. proposed a plug-and-play framework that encodes the KB schema into the parameters of a LoRA Hu et al. (2022) module. But parameterizing the KB may introduce extra errors and result in a loss of interpretability.

We follow the second line of work, aiming to address the challenge of transferability, interpretability and accuracy at the same time.

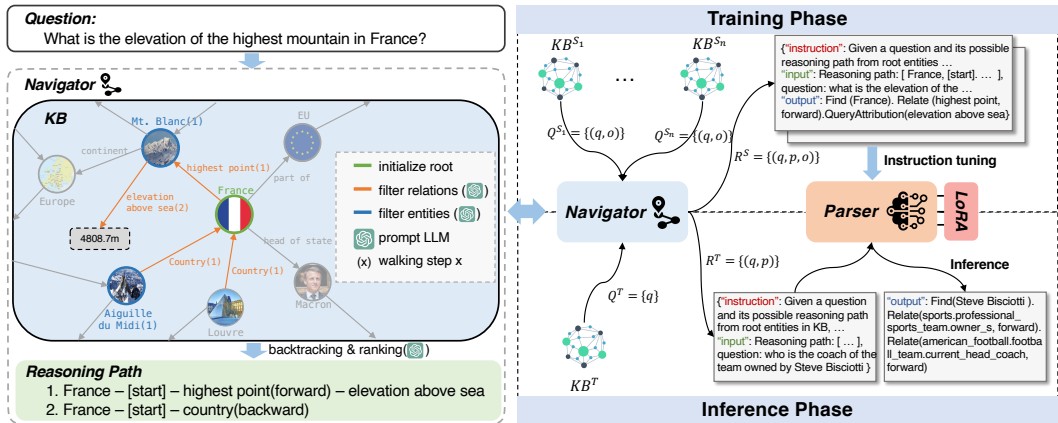

Figure 2: An illustration of the training and inference of REPANA framework.

# 3 PRELIMINARY

In this section, we introduce the formal definition of the knowledge base (KB) and then formulate our task on KB.

**Knowledge Base (KB)**. A knowledge base can be formally described by $\mathcal{G} = \{\mathcal{E}, \mathcal{C}, \mathcal{R}, \mathcal{T}\}$, where $\mathcal{E}$, $\mathcal{C}$, $\mathcal{R}$ and $\mathcal{T}$ denote the set of entities, concepts, relations and triples, respectively. Each entity $e \in \mathcal{E}$ is assigned a unique ID and belongs to one or more concept $c \in \mathcal{C}$. $\mathcal{R}$ contains the special relation $r_e =$"instanceOf", $r_c =$ "subClassOf" and the general relation set $R_l = r_l$. Given $\mathcal{E}, \mathcal{C}$ and $\mathcal{R}$, $\mathcal{T}$ can be divided into three subsets: (1) "instanceOf" triple set $\mathcal{T}_e = \{(e, r_e, c) | e \in \mathcal{E}, c \in \mathcal{C}\}$ ; (2) "subClassOf" triple set $\mathcal{T}_c = \{(c_i, r_c, c_j) | c_i, c_i \in \mathcal{C}\}$; (3) general relation set $\mathcal{T}_l = \{(e_i, r_l, e_j) | e_i, e_j \in \mathcal{E}\}$.

**Program**. As stated before, we choose KoPL as the program language, for it is well modularized and LLM-friendly. KoPL is composed of symbolic functions with arguments arranged in the tree structure. Each function defines a fundamental operation in KB. This tree can be serialize with post-order traversal into $y = \langle f_1(arg_1), \cdots, f_i(arg_i), \cdots, f_{|y|}(arg_{|y|}) \rangle$ where $f_i \in \mathcal{F}, arg_i \in \mathcal{E} \cup \mathcal{C} \cup \mathcal{R}_l \cup \{\emptyset\}$

**Problem Formulation.** In this work, we assume that the KB is available and there are one or more root entities in the given question. We further assume that there exists the answer to the question and a viable reasoning path from the root entity to the answer. Formally, given a KB $\mathcal{G}$ and a natural language question $x$ with its root entity $\{e_1, \cdots, e_m\}$, we aim to first retrieve the corresponding reasoning path $p = \{\langle e_1, r_{11}, \cdots, r_{1k_1}\rangle, \cdots, \langle e_m, r_{m1}, \cdots, r_{mk_2}\rangle\}$, where $r_{ij} \in \mathcal{R}, k_1, k_2 \leq k$ - the maximum path length. Then use $x$ along with $p$ as a navigation to generate the program $y$, which would return the correct answer.

# 4 FRAMEWORK

In this section, we introduce the main components of our reasoning path navigation framework and how they work together.

First, we want to give an overview of the framework. As mentioned in the introduction, we face two major challenges in implementing the system: (1) how to make sure the knowledge retrieving is accurate and concise, while applicable to all KBs; (2) how to ensure the parser does not over fit to one KB's schema. To address these two problems, we introduce our reasoning path navigated program induction framework, containing the **KB navigator** with KB-walk search strategy and the **KB-agnostic parser** with denoising mixed instruction tuning strategy, shown in Figure 2.

The framework generally follows the two-stage retrieve-and-generate paradigm. In the training phase, we first employ the KB navigator module to extract the reasoning path $p$ of the input question $q$ from the corresponding KB. Then we gather all the questions $Q^S = \{Q^{S_1}, Q^{S_2}, \cdots, Q^{S_n}\}$ from

$n$ expanded KBs $KB^S = \{KB^{S_1}, KB^{S_2}, \cdots, KB^{S_n}\}$ and their corresponding reasoning path $p$ to construct an instruction dataset $R^S = \{(q, p, o)\}$, where $o$ is the output program. After instruction tuning the KB-agnostic parser using the mixed dataset, it is ready to inference on the target low-resource $KB^T$. Similar to the training phase, the framework also need to first retrieve the reasoning path $p$ from the target $KB^T$, and then feed both the input question $q$ and its retrieved path $p$ with instructions to the parser, which will finally output the program executable on the target $KB^T$.

In the following we will introduce the details of the implementation of the main components of our framework: KB Navigator (Section 4.1) and KB-agnostic parser (Section 4.2). We will also introduce other modules that play a part in the framework (Section 4.2.1).

## 4.1 KB Navigator

Given a question, the KB navigator leverage its underlying KB to localize the corresponding reasoning paths. We propose the KB-walk search strategy based on two observations: (1) despite schema differeces between KBs, all KBs are constructed with knowledge elements such as entity, relation and concep, and are organized as a graph. So it is plausible to perform a walk algorithm on the graph in all KBs. (2) LLMs are extremely good at selecting the correct relations relate to the question from a bunch of candidates without further fine-tuning, which is suitable for navigation.

### 4.1.1 Reasoning Path Construction

Section 3 has given a general description of KB. Based on it, here we define four groups of knowledge elements in KBs: entity, concept, relation, qualifier. Entity, concept and relation is the same as the general description, only that the "relation" contains both relations between entities (e.g., part of) and attributes between a entity and a value (i.e., population), which in this paper we uniformly refer to it as relation. Qualifier is the extra description related to the triple in some KBs, e.g., ((France, part of, EU), start time, 1957).

In the construction of our reasoning path, we take the entity $e$ and relation $r$ to form the main structure of the path. A reasoning path can be generally denote as $p = \langle e_r, [start], r_1, \cdots, r_k \rangle$, where $e_r$ represents the root entity of the path, $k$ is the walking range. Additionally, the concept $c$ and qualifier $u$ are also append to path $p$ as an extra list for the convenience of parsing. Noted that there might multiple root entities in a question, in which case, the KB navigator will return more than one path, each corresponding to one root entity. Some paths may partially overlap, and the understanding of the paths is taken into next step of parsing.

### 4.1.2 KB-walk Search Process

The process of KB-walk contains the following 4 steps: **initialize, filter relations, filter entities, backtrack & rank**. The 2nd and the 3rd step will be repeated $k$ rounds. $k$ is maximum walk range.

**Initialize.** In this step, KB navigator mainly initialize the root entities $\{e_r\}_{r=1,2,\cdots,m}$ of the search algorithm. We use the topic entities of the input question as the root entities, which is often provided by the dataset. Existing off-the-shelf named entity recognition models can also satisfy the need, which is not the main focus in this work.

**Filter relations.** This step aims to explore the surroundings of the given start nodes, and to select suitable directions for advancement from the rooot nodes in each of the total $k$ rounds of traversal. Therefore, there are two main actions in this step:

- **Query.** In the $i$-th round, the start entities are denoted as $E_i = \{e_{1,i}, e_{2,i}, \cdots, e_{b,i}\}$, where $b = m$ when $i = 1$ else $b$ equals beam size. We query the KB and gather all the relations $\hat{R} = \{(r_{1,1}, r_{1,2}, \cdots), \cdots, (r_{b,1}, r_{b,2}, \cdots)\}$ that connects the each entity in $E_i$ both inwardly and outwardly. In Figure 2, $E_1 = \{$ France $\}$, $R_1 = \{$ highest point, part of, head of state, country $\}$.

- **Filter.** After the $R_i$ is gathered, we prompt the LLM to choose up to $b$ relations from $R_i$ (could be 'no answer') given the question and $E_i$, and get $F_i = \{r_1, r_2, \cdots, r_b\}$. In the case of Figure 2, $F_i = \{$ highest point, country $\}$.

**Filter entities.** This step aims to take a step forward along $F_i$, walk onto the target entity, and then filter them and form $E_{i+1}$ as the start nodes of $i + 1$ round. There also are two action:

- **Query.** In the $i$-th round, we walk from $E_i$ along $F_i = \{r_1, r_2, \cdots, r_b\}$, which yield $b$ beams of target entities $\hat{E} = \{(e_{1,1}, e_{1,2}, \cdots), \cdots, (e_{b,1}, e_{b,2}, \cdots)\}$. In Figure 2, the $\hat{E} = \{(\text{Mt. Blanc}),(\text{Louvre, Aiguille du Midi})\}$.

- **Filter.** We need to select one entity from each of the beam buckets to get $E_{i+1}$. We can prompt the LLM multiple times to get the answer, but in practice, considering the cost, we randomly select one entity from each buckets, assuming that entities in one buckets are of the same type and share similar relations. Since there is no intermediate entity in the reasoning path, we find it works fine in our framework. In Figure 2, $E_{i+1} = \{\text{Mt. Blanc}\}$.

**Backtrack & rank**. In the final step, we backtrack the path to the root entity and collect the path of all lengths as candidates, and them prompt the LLM to rank the path based on relevance to the question. Noted that the relations in the path are tagged with their original **direction**. In the case of Figure 2, there are two candidates and LLM gives a rank.

## 4.2 KB-AGNOSTIC PARSER

To avoid over fitting the parser to a single KB schema, making it difficult to transfer to other question datasets built on different KBs, we employ the denoising instruction tuning with the reasoning path as part of the input. Since the reasoning path may contains a small amount of noise, such as omission of some schema items, the parser has to denoise from the input to construct the program.

As introduced above, we gather questions from multiple questions from the rich resource KB and retrieve their reasoning path to construct an dataset $R^s = \{(q, p, o)\}$, where $o$ is the output program. To construct the intruction tuning dataset, we first convert the entity IDs (e.g., m.0f8l9c) into its friendly names (e.g., France). Then we standardize this data into a unified format, where $q$ and $p$ are put into "input" tag and $o$ are "output" tag, as shown in Figure 2. The "instruction" is unified across the training and testing dataset. To increase the diversity of the training set, we also paraphrase the training set into $n$ expanded sets. Not only the input question, but also the schema items in the output program are paraphrased. For example, the relation "Highest poing" may be paraphrased into "Peak elevation". In this way, we expand the original KB into $n$ variations $KB^S = \{KB^{S_1}, KB^{S_2}, \cdots, KB^{S_n}\}$. Through the denoising mixed instruction tuning, the the parser is expected to focus more on program's sketches (i.e., the function names and their structure), generate the function's argument will be more like a selection and completion task.

### 4.2.1 PARAMETER EFFICIENT FINE-TUNING

REPANA also adopts the parameter efficient fine-tuning technique with LoRA (Hu et al., 2022), a popular type of expandable module for LLMs with fewer trainable parameters. Specifically, LoRA adds an extra forward pass to the specified matrix $W_i \in \mathbb{R}^{m \times n}$ within the LLM, changing the original pass $h = W_i x$ into $h = (W_i + A_i B_i)x$, where $A_i \in \mathbb{R}^{m \times r}, B_i \in \mathbb{R}^{r \times n}, r \ll \min(m, n)$. During training, the original parameter $W_i$ is frozen and only $A_i, B_i$ is trainable. In this way, REPANA is able to reduce training costs while using larger LLMs as the backbone model.

## 4.3 POST-VALIDATION MODULES

In this section, we briefly describe post-validation modules in the framework, which is consist of the direction check and relation check. In the experiment, we observe that the parser is particularly insensitive to the direction of relation in the reasoning path, even the directions are already indicated after the relations in brackets. To solve this problem, we leverage a rule-based correction module, where the final program undergoes verification based on the direction of the same relations contained in the reasoning path of the question. We found that this strategy alone can significantly improves the accuracy of the final model. Additionally, due to the possible absence of schema items in the reasoning path, the model sometimes generate a similar label based on the training data. In this case, we substitute the label with the most similar label in the target KB.

## 5 EXPERIMENTS

### 5.1 DATASETS

**Rich-resource Dataset.** KQA Pro (Cao et al., 2022a) built on Wikidata is a popular and well-annotated rich-resource KBQA dataset. We sample questions from it to construct a 60k training set, ensuring that there is at least one topic entity in the question.

**Low-resource Dataset.** Apart from KQA Pro, we adopt GrailQA (Gu et al., 2021), WebQuestions Semantic Parses(WebQSP) Yih et al. (2016), ComplexWebQuestions (ComplexWQ) (Talmor & Berant, 2018) and MetaQA (Zhang et al., 2018) as the target low-resource datasets. The first three datasets are built on Freebase, another popular KB. For MetaQA, it is built on WikiMovies in the domain of movies. So it can evaluate our framework's transferability to specific domains in detail. In addition, it is divided into three subsets by the reasoning hops, making it convenient to study performance in single-hop and multi-hop scenarios. Since most relation in MetaQA's KB are covered by KQA Pro, we remove certain data entry to make sure that these schema items is not included in the KQA Pro training set. Overall, almost all schema items in the target datasets are unseen in the source datasets. We use the test questions of KQA Pro validate if REPANA can well generalize on the mixed training data, and use the test question from the latter four aims to validate the transferability.

### 5.2 BASELINES

In this section, we mainly introduce the supervised and low-resource PI methods for the WebQSP, CWQ and MetaQA.

The supervised models include: (1) **PullNet** (Sun et al., 2019) proposes to iteratively construct a subgraph from KB and text for effective multi-hop reasoning; (2) **TransferNet** (Shi et al., 2021) presents a model that incorporates transparent graph searching and attention-based method to perform interpretable reasoning. (3) **RnG-KBQA** (Ye et al., 2022) introduces a retrieve-and-generate framework that enumerates and ranks all relevant paths for program generation. (4) **ChatKBQA** (Luo et al., 2023) presents an instruction tuning method for LLMs, which perform PI by first generating and then grounding labels to the KB. (5) **KG-Agent** (Jiang et al., 2024) introduces an LLM agent that is able to explore the KB with a set of pre-defined tools and performs a step-by-step reasoning by asking the LLM to take appropriate actions based on the history information.

The low-resource methods are as follows: (1) **StructGPT** (Jiang et al., 2023a) can be regarded as an early version of KG-Agent with fewer operations, but it has a wider range of applicability and does not require training data. (2) **ToG** (Sun et al., 2024) proposed a explore-and-think strategy based on the knowledge graph, starting from the topic entity, leverage LLM to select relevant relations and reason on it. (3) **KB-Binder** (Li et al., 2023a) first proposed to utilize the in-context learning ability of LLMs to generate program with a few question-program examples provided in the prompt. (3) **Pangu** (Gu et al., 2023) introduces an PI method that utilize the LLM to rank the candidates in the process of rule-based program expansion with in-context learning. (4) **ProgramTrans** (Cao et al., 2022b) is the first to propose the program transfer paradigm for low-resource scenarios, leveraging a two-stage generation framework with an ontology-guided pruning strategy. (5) **KB-Plugin** (Zhang et al., 2024) presents a method that encodes the KB schema into the model's parameters to build a plug-and-play framework for low-resource KBs.

### 5.3 METRICS

Following prior works (Cao et al., 2022a; Zhang et al., 2024; Jiang et al., 2024), we use F1 score for GrailQA, WebQSP and CWQ, and use Hit@1 for MetaQA, and accuracy for KQA Pro.

### 5.4 IMPLEMENTATION

In experiments, we use the Llama-2-7B (Touvron et al., 2023) and Meta-Llama-3-8B-Instruct (Meta, 2024) as the backbone LLM to train the parser. The parameter of LoRA is set to $r = 8, \alpha = 32$ during training. With respect to the KB navigator, we use ChatGPT-3.5-turbo (OpenAI, 2024a) as the navigation LLM and set the beam size to 5 and walk range to 3. We utilize $4 \times$A100 GPUs to

train the parser for 5 epochs with learning rate $1e - 4$, batch size $64$, gradient accumulation 2 and weight decay $0.01$. All the prompts used in the framework can be found in Appendix B.

| Model | GrailQA | WebQSP | ComplexWQ | MetaQA | | |
| --- | --- | --- | --- | --- | --- | --- |
| | | | | 1-hop | 2-hop | 3-hop |
| *Supervised* | | | | | | |
| PullNet | - | 62.8 | - | 97.0 | 99.9 | 91.4 |
| Transfernet | - | - | - | 97.5 | 100.0 | 100.0 |
| RnG-KBQA | 76.9 | 75.6 | - | - | - | - |
| ChatKBQA | - | 79.8 | 77.8 | - | - | - |
| KG-Agent | 86.1 | 81.0 | 69.8 | 97.1 | 98.0 | 92.1 |
| *Low-resource* | | | | | | |
| ProgramTrans$_\dagger$ | - | 53.8 | 45.9 | - | - | - |
| KB-Binder(6 shots) | 56.0 | 53.2 | - | 93.5 | 99.6 | 96.4 |
| KB-Plugin | 65.0 | 61.1 | - | 97.1 | 100.0 | **99.3** |
| Pangu(100 shots) | 62.7 | 68.3 | - | - | - | - |
| StructGPT$_\dagger$ | - | 69.6 | - | **97.1** | 97.3 | 87.0 |
| ToG(w/ ChatGPT) | 68.7 | 76.2 | 57.1 | - | - | - |
| ours(Llama2-7B)$_\dagger$ | 78.6 | 76.7 | 51.5 | 94.6 | 100.0 | 95.1 |
| -w/o DC | 64.2 | 58.6 | 26.3 | 89.3 | 94.6 | 90.5 |
| ours(Llama3-8B)$_\dagger$ | **81.3** | **79.2** | **57.6** | 96.2 | **100.0** | 97.0 |

Table 1: F1 results on GrailQA, WebQSP and ComplexWQ. Hits@1 results on MetaQA. The $\dagger$ means the method uses the oracle topic entities. DC means direction correcting. For all low-resource baselines, we report their results without using any parallel data from the target dataset.

# 6 RESULTS

## 6.1 MAIN RESULTS

In this work, we focus on the transferability on the low-resource KB. Therefore, we mainly compare REPANA with low-resource methods. The results are presented in Table 1 and 2.

In Table 1, the three datasets are all unseen during training. For GrailQA and WebQSP, REPANA outperforms most low-resource methods by a large margin, despite the models like StructGPT and Pangu using much larger backbone models, and is even comparable to some supervised methods. This indicates that REPANA performs excellently on question with fewer inference hops in WebQSP. We believe this is because REPANA can accurately provide paths in the target KB that include the correct relations, allowing the parser to select from these and generate correct

| | Model | Accuracy |
| --- | --- | --- |
| *Supervised* | RGCN (Schlichtkrull et al., 2018) | 35.1 |
| | BART+KoPL (Cao et al., 2022a) | 90.6 |
| | CFQ IR (Herzig et al., 2021) | 89.0 |
| | GraphQ IR (Nie et al., 2022) | 91.7 |
| | KG-Agent | **92.2** |
| | Ours* | 92.0 |
| *Low-resource* | Fine-tuning | 22.5 |
| | LLM-ICL | 31.8 |
| | FlexKBQA (Li et al., 2024) | 46.9 |

Table 2: Accuracy on KQA Pro. * is result of dev set.

programs. On the more difficult CWQ dataset with more hops, REPANA's performance only exceeds ToG by 0.5%. In our observations, we found that REPANA's path navigation is prone to errors in questions with longer inference chains, leading to much lower performance comparing to supervised methods. Regarding MetaQA, since its KB is relatively small, most recent low-resource methods have achieved or even surpassed supervised methods, and REPANA has also reached the level of SoTA. We noticed that all methods perform worse on 1-hop set compared to multi-hop sets. For REPANA, it is because the 1-hop dataset includes "tag_to_movie" types, involving lookup of entities from attributes. REPANA currently cannot handle such questions that lack a topic entity, resulting in relatively lower performance.

Table 2 presents the result on KQA Pro. Since our parser is trained on the mix of paraphrased datasets, we put REPANA into the supervised group. But since we excluded questions that has schema overlap with the training set during testing, it can actually serve as a zero-shot experiment on unseen KB schema items. The results indicate that REPANA's performance on KQA Pro is comparable to supervised SoTA. Considering the fact that we did not use the complete training set, and the noise introducing in the denoising mixed training, we can safely conclude that, overall, REPANA generalizes well on the paraphrased mixed heterogeneous training set.

## 6.2 ABLATION STUDY

### 6.2.1 MIXED TRAINING EFFECTIVENESS EVALUATION

To evaluate the effectiveness of the proposed mixed instruction tuning strategy, we compare REPANA that trained on the original KQA Pro, and a different number of the mixed variations of the original dataset with the Llama-2-7B as the backbone model for the parser.

On one hand, the results in Table 3 indicate that even without paraphrasing the original KQA Pro into a mix of variations of datasets, RENAPA with only the help of input reasoning path can already achieve 64.9 F1 score on WebQSP, which is comparable to many low-resource method such as KB-Plugin and Pangu. On the other hand, the increase of the different variation of the original KQA Pro dataset can indeed improve the performance on the task of transferring to low-resource heterogeneous data. Integrating three paraphrased variations with original KQA Pro dataset results in a 7% improvement in performance, validating the effectiveness of mixed training. Based on this, we can reasonably speculate that incorporating more heterogeneous training data would further enhance the model's transfer capabilities.

| Model | WebQSP | CWQ |
|---|---|---|
| REPANA$_{kqapro}$ | 69.5 | 45.6 |
| REPANA$_{mixed-2}$ | 72.4 | 49.1 |
| REPANA$_{mixed-3}$ | 75.5 | 50.4 |
| REPANA$_{mixed-4}$ | 76.7 | 51.5 |

Table 3: Ablation on the effectiveness of mixed instruction tuning. $kqapro$, $grailqa$ and $mixed$ represents the model trained on KQA Pro only, GrailQA only and the mixed training set, respectively.

### 6.2.2 REASONING PATH EFFECTIVENESS EVALUATION

To validate the importance of the structure of reasoning path as part of the input, we compare parsers that trained with three input of KB information: (1) gold program and reasoning path; (2) gold program and lists of schema items (entity, relation, concept, qualifier) (3) gold program only.

Results in Table 4 shows that apart from the accurate names of schema items in the target KB, the structures included in the reasoning paths are also crucial for the performance of transferability. If the input only includes the relevant schema items but lacks their structural information, the model will struggle to organize them correctly, resulting in a performance drop of more than half. Moreover, the parser learning the KB schema solely from program-question pairs from the training set clearly cannot transfer to other heterogeneous KBs.

| Model | WebQSP | CWQ |
|---|---|---|
| REPANA$_{none}$ | 12.6 | 5.3 |
| REPANA$_{list}$ | 43.1 | 23.9 |
| REPANA$_{path}$ | 76.7 | 51.5 |

Table 4: Ablation on the effectiveness of reason path. $none$, $list$ and $path$ means the input of no KB info, lists, and path.

### 6.2.3 LLMs NAVIGATION EVALUATION

In this section we validate the basic observation that LLMs are very skilled at selecting the correct relations relate to the question without further fine-tuning. We evaluate ChatGPT-3.5-turbo (OpenAI, 2024a), GPT-4o (OpenAI, 2024b), GLM-3-Turbo (ThuDM, 2024) and GLM-4-9B on 100 one-hop questions sampled from GrailQA.

In the experiment, we ask LLM to choose $K$ ($K = \{1, \cdots, 5\}$) relations from the list of candidates, and record the recall score in the top-$K$ result (Hit@$K$). We run the experiment for three times and results are shown in Figure 3.

Note that here $K$ is equivalent to the beam size in our algorithm. The results show that these LLMs perform well on this task under zero-shot conditions, considering that Freebase is quite dense and contains many similar relations. Especially, GLM-4-9B and GPT-4o are on par, both achieving a recall rate of over 90% when the beam size is set to 5.

# 7 CONCLUSION

In this paper, we propose REPANA, a reasoning path navigated framework that enables LLMs to universally perform reasoning on low-resource datasets by providing the KB-agnostic parser with the reasoning paths in target KBs with the help of the novel KB navigator. REPANA achieves better performance on the four heterogeneous target datasets with much smaller backbone models compared to

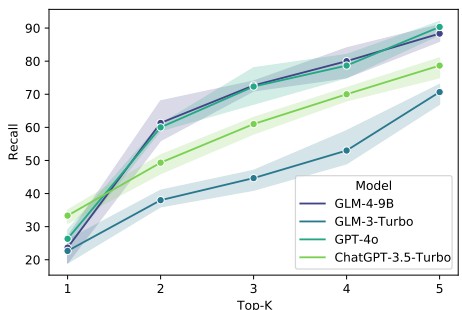

Figure 3: Popular LLMs' zero-shot performance of selecting the one-hop relation based on the given question.

other low-resource PI methods, even on par with some supervised methods. The ablation studies further validate the effectiveness of our proposed KB-walk retrieving strategy and mixed instruction tuning in low-resource scenarios. Although there are limitations that the proposed retrieving algorithm also relies on the topic entity, and searching accuracy may drop with the increase of the hops of question, we plan to address these issues in the future work.

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

## A    APPENDIX

## B    USED PROMPTS IN EXPERIMENT

All the prompts used in the framework are shown in Table 5. To reduce cost, the path ranking prompt is replaced with random selection in practice as mentioned in the paper.

## C    ERROR ANALYSIS

In this section we analyze the main types of error of REPANA. As shown in Table 6, we categorized them into the following groups:

- **Relation direction error.** This is the most common error in experiment. The parser tend to overlook the direction of relation given in the path, and generate wrong direction. However, it is an easy problem. As mentioned in the paper, we use a rule-based correction module to revise the generated program according to the retrieved path.

- **Long path ranking error.** When the hops of the question increases, the length of the path goes longer, and it is more likely to result in errors in one of the searching steps. And when the path gets longer, there are similar paths in candidate, or the path start from one topic

| Functionality | Prompt |
|---|---|
| *Filter relations* | In order to answer the question "%s", from the relations of relevant entities %s, select the top %s relations that are most helpful to answer the question: [%s]. Just answer the names. |
| *Filter entities* | From the entity list: [%s] that maybe relevant to the question '%s', select the top %s entity that are most helpful to answer the question. Just answer the names. |
| *Path ranking* | From the given list of relation paths in the knowledge base, select the top %s paths that are most relevant to the knowledge required to answer question %s.
The paths are: [%s], answer the complete path. |
| *Training instruction* | ### Instruction: Given a question and its possible reasoning path from root entities in knowledge base, generate a Logical Form query according to the question.
Input: Reasoning paths: [%s]. Other elements - concept: [%s], qualifier: [%s]. Question: %s.
### Output: %s ### |

Table 5: The used prompts and instruction of the framework. %s means the corresponding content.

entity of the question to another topic entity instead of the answer. In both situations, it is difficult for LLM to distinguish the differences and could make mistakes. The example in Table 6 shows the second situation, where the correct path contains two branches, each one is from entity (goiás, bolivia) to answer (Brazil). But in the retrieved path, the red relations are repeated, leading the path from the goiás to bolivia and bolivia to goiás.

- **Multi-hop generation error.** We find that sometimes when the retrieved path is correct, let's say a 3-hop path, but the parser neglects the last step of the path, only generate the first two hops. This error is probably related to the last ranking error, due to the path mistake in the training set, resulting in the mismatch between the input path and gold program.

- **Program sketch induction error.** This error is another common error. It happens when the question and program are very complex, e.g., multiple topic entities and long reasoning path. This problem is probably because of the training data. Since we only use 30k pairs from KQA Pro and GrailQA, and the complex question is rare, especially in GrailQA. Also, the correct reasoning path of complex question is difficult to retrieve, so there is a large chance of mismatching between path and program.

| Error Type | Example |
|---|---|
| *Relation direction* | **Question**: what does jamaican people speak?
**Path**: [ jamaican, [start], location.country.languages_spoken(forward) ]
**Output**: Find(jamaican).Relate(location.country.languages_spoken, backward. what() |
| *Long path ranking* | **Question**: what does bolivia border and is the country that contains goiás?
**Gold program**: Find(goiás).Relate(location.country.administrative_divisions, backward).Find(bolivia).Relate(location.location.adjoin_s, forward).Relate(location.adjoining_relationship.adjoins, forward).And().What()
**Gold path**: [[goiás, [start], location.country.administrative_divisions (backward)], [bolivia, location.location.adjoin_s(forward), location. adjoining_relationship.adjoins(forward)]]
**Retrieved path**: [[ goiás, [start], location.country.administrative_divisions (backward), location.location.adjoin_s(forward), location. adjoining_relationship.adjoins(forward) ], [bolivia, [start], location.location.adjoin_s(forward), location. adjoining_relationship.adjoins(forward), location.country.administrative _divisions(forward)]] |
| *Multi-hop generation* | **Question**: who is listed as screenwriter of the movies starred by My Big Fat Greek Wedding actors?
**Path**: [my big fat greek wedding, [start], starred_actors(forward), starred_actors(backward), written_by(forward)]
**Output**: Find(My Big Fat Greek Wedding).Relate(starred_actors, forward) Relate(starred_actors, backward).What() (Missing written_by) |
| Sketch induction | **Question**: What is the hometown of the architect who designed mount vernon?
**Path**: [mount vernon, [start], architecture.architect.structures_designed(backward) people.person.place_of_birth(forward)]
**Output**: Find(mount vernon).Relate(architecture.architect.structures_designed, backward).QueryAttr(people.person.place_of_birth) |

Table 6: Error types and examples.

