# OpenReview forum: "REPANA: Reasoning Path Navigated Program Induction for Universally Reasoning over Heterogeneous Knowledge Bases"
_ICLR.cc/2025/Conference — Submitted to ICLR 2025_

### Official Review · Reviewer_2g3F · 2024-10-30

**Soundness:** 3
**Presentation:** 3
**Contribution:** 2
**Rating:** 5
**Confidence:** 3

**Summary:**

The paper proposed a new approach for KB based reasoning task. The approach consists two steps: 1) KB navigator to collect relevant paths (schema information) from KB. 2) A KB-agnostic parser to generate the program. The KB navigator leverages the underlying KB to localize the corresponding reasoning paths using the walk-search process. The parser is trained using LoRA and is focusing on learning sketches of the output program. The parser is trained with multiple heterogeneous datasets to maximize the generalization across different KB databases. The paper shows decent performance boost on unseen low-resource databases such as GrailQA, WebQSP, ComplexWQ coming with existing methods. In the ablation studies, it further validates the effectiveness of the proposed KB-walk retrieving strategy and mixed instruction training for low-resource scenarios.

**Strengths:**

1. The paper presents a framework for KB based reasoning task. It tackles one of the challenges with KB QA that different knowledge graph has different schema for program induction. The authors set up complicated KB pipeline to investigate how to improve the generalization capability of such system.
2. Evaluation is comprehensive. The authors includes many of the previous work and conduct thorough experiments to demonstrated that the proposed approach helps on low-resources settings. Ablation studies seems solid. The performance boost is decent comparing with other approaches.
3. Although there are some grammatical issues in the writing, the paper in general is easy to follow and well-structured.

**Weaknesses:**

1. It would be great if the authors could pay more attention to the writing. Here are some grammatical issues:
 - Line 108, fist -> first
- Line 228, leverage -> leverages
- Line 289, Over fitting -> overfitting
- Line 323, generate -> generates
- Line 114, missing on at the end of the line
- Line 437, introducing -> introduced
2. For the KB parser, the proposed approach leverages a rephrasing approach to make output program heterogenous, but fails to mention how to recover the entities and relations at inference time. Would that be possible to shed light on how this is done?
3. The proposed method involves an iterative way to extract a subgraph of a KB and LoRA finetune an LLM to generate a homogenous output program. Given that there are already some research works on unifying the output program (e.g. [1] here), I fail to see the main contribution from the proposed approach.

[1] XSemPLR: Cross-lingual semantic parsing in multiple natural languages and meaning representations

**Questions:**

1. Because the parser training masks the KB specific relations, how do we recover such relations at inference time? For instance, in Line `the relation “Highest poing"` (is it a typo here?) will be rephrased to "peak elevation", how do we map it back at inference time? Have you conducted any experiments to check the accuracy of such mapping?

---

> ### Author Response · Authors · 2024-11-21
> **Reply**
>
> Your suggestions are greatly appreciated. We reorganize your questions as follows:
>
> Q1: Regarding writing issue.
>
> We apologize for the typos and mistakes made in a hurry. We will do our best to improve the writing in the next version.
>
> Q2: Regarding rephrase mapping.
>
> We are afraid there is a misunderstanding. The paraphrasing is only used in training, not in inference, and there is no need to map it back. For example, question: Height of Mt. Everest?, the original input-output pair is:
>
> “‘Input’: { ‘Question’: ‘Height of Mt. Fuji?’, ‘Reasoning path’: [Mt. Fuji, highest point] }, ‘Output’: ‘Find(Mt. Fuji).relate(highest point)’”
>
> , a paraphrase version would be:
>
> “‘Input’: { ‘Question’: ‘Height of Mt. Fuji?’, ‘Reasoning path’: [Mt. Fuji, peak elevation] }, ‘Output’: ‘Find(Mt. Fuji).relate(peak elevation)’”
>
> The parser only learns the mapping from the input to output string in training.
> And there is no paraphrasing in inference, so there is no need for mapping back.
>
> Q3: Regarding our contribution comparing to other works on unifying the output program.
>
> The main contribution of this work is not in how we unify the program language to answer questions from different datasets based on different knowledge bases (KBs). Rather, this work aims to address the challenge that a parser trained on one KB is often difficult to transfer to another. In this context, we introduce the reasoning path, which is retrieved by LLMs through iterative exploration of the KB, to augment the input and provide KB schema information. From our perspective, the key contribution is proposing a framework that decouples KB schema perception from program generation, thus allowing for greater transferability across different KBs and reducing the need for extensive manual annotation.

---

> ### Author Response · Authors · 2024-11-29
> **Follow-up on Rebuttal Submission**
>
> Dear Reviewer,
>
> I hope you had a wonderful and restful Thanksgiving holiday. I want to follow up on my rebuttal submission for this paper. I truly value the time and effort you’ve put into reviewing my work, and I hope my responses have addressed your concerns and questions.
>
> If you’ve had the opportunity to go through the rebuttal, I would be grateful if you could consider the additional clarifications and improvements I’ve provided. If feasible, I would sincerely appreciate it if you could revisit your score in light of these updates.
>
> Thank you once again for your time and thoughtful review.

---

> > ### Comment · Reviewer_2g3F · 2024-12-03
> >
> > Thank you for the clarifications. I see your point regarding to the output mapping. My follow-up question is would that limit the capability of the generated program? I would assume the generated programs still have to apply to specific KBs.

---

> > > ### Author Response · Authors · 2024-12-03
> > > **Reply to the follow-up question**
> > >
> > > Thank you for your time and effort.
> > >
> > > Regarding your question, we believe the paraphrasing does not affect the capability of the generated program based on two reasons:
> > >
> > > 1. The capability of a program (we assume it means the types of questions the program can answer) primarily depends on the combination of operators. The paraphrasing operation only affects the arguments within the operators and does not alter their compositionality. Thus, the questions that could be answered before can still be answered after the paraphrasing.
> > >
> > > 2. Paraphrasing the labels of a KB can be regarded as creating a new KB. The structure of the new KB remains identical to the original, with only differences in the schema naming. It can be considered that the generated program actually grounds to this new KB. If we just maintain a dictionary of the original label names mapping to the paraphrased names, it won't be a problem to apply the generated programs to the new KB.
> > >
> > > We hope this explanation has addressed your concerns. Thank you once again for your attention and consideration.

---

### Official Review · Reviewer_pGPC · 2024-11-02

**Soundness:** 3
**Presentation:** 1
**Contribution:** 3
**Rating:** 6
**Confidence:** 2

**Summary:**

This paper addresses the problem of answering queries using a knowledge base (KB). The paper proposes a system in which a LM is conditioned on the given
query and generates a reasoning path through the KB. Then the answer to the query can be obtained by executing the path on the KB. The system consists of two
parts: a navigator and a KB-agnostic parser, both of which are based on LMs. The navigator proposes a reasoning path based on the question, and the parser takes
as input the question and proposed reasoning path and outputs a final program. The parser is KB-agnostic because it is trained across a variety of KBs and can
be applied to a new KB without extra training. Compared to other methods that do not train on the target KB, the proposed method obtains better F1. An ablation
shows that the reasoning path input to the parser is a crucial part of the system when evaluated on unseen KBs.

**Strengths:**

- The paper proposes a method for the knowledge-base reasoning problem
- The results of the proposed method are better than those of other "low-resource" methods
- Table 4 shows the results of an ablation indicating that the reasoning path is crucial for good performance on unseen KBs.

**Weaknesses:**

The writing has many shortcomings:
- The method section is somewhat unclear to me. For example, is the process in 4.1.2 used only for inference, or is it used somehow in training or to create the training data?
- Continuning on the previous point, what is the difference between a "reasoning path" (ostensibly the output of the navigator) and a "program" (ostensibly the
  output of the parser)?
- Line 54 says that a shortcoming of agent methods is that they require a "topic entity" to occur in the question, but from line 256 ("We use the topic entities of the input question"), it seems REPANA has the same weakness
- There are many typos / grammatical errors (please see the "Questions" section for some examples)

**Questions:**

Typos:
- line 50: "still rely" -> "still relies"
- line 104: "Through of this naval" -> "Through this novel"
- line 108: "fist train" -> "first train"
- line 260: "rooot" -> "root"
- line 321: "can significantly improves" -> "can significantly improve"
- line 323: "model sometimes generate" -> "model sometimes generates"
- line 339: "almost all schema items .. are unseen" - can you quantify this?
- Table 2: why is REPANA's result marked as dev set while all others are not?
- Table 3: grailqa is mentioned but not included in the table

---

> ### Author Response · Authors · 2024-11-21
> **Reply**
>
> Thank you for your valuable suggestions, we reorganize your concerns as follows:
>
> Q1: Regarding writing issue.
>
> We apologize for the typos and mistakes made while writing in a hurry. We will do our best to improve the writing in the next version.
>
> Q2: Regarding the process in 4.1.2.
>
> Sorry for not making it clearer in the paper. The process in Section 4.1.2, which retrieves a reasoning path from the KB for a question, is used for both training and inference. As demonstrated in Figure 2, the retrieved path is part of the input and is in a consistent format during both training and inference.
>
> Q3: Difference between a reasoning path and a program.
>
> The reasoning path consists of the relations along the path from the starting root entity to the answer in the knowledge graph, which mainly includes the names of relations. A program, on the other hand, is a complete query of a formal language, such as SPARQL or KoPL, with their defined identifiers, symbols, and grammar rules. In general, an oracle reasoning path is a subset of the argument set of the program, which represents the exact information from the knowledge base that LLMs need to generate the program.
>
> For example, the program Find(France).relate(president) means first find the entity "France", and then relate it to its "president". Its oracle path is [France, <start>, president].
>
> Q4: Regarding the weakness of needing topic entity.
>
> Sorry for the writing issue again. We don’t intend to emphasize that our approach is superior just because we overcome this weakness. In fact, you are correct that the current version of REPANA also suffers from this weakness, and we have mentioned this limitation in the conclusion. However, we believe we have two advantages: (1) We can easily improve the first retrieval stage without changing the framework, as we have planned and mentioned in the conclusion. (2) The failure of the first stage won’t stop the reasoning process, unlike previous approaches. The second stage has the opportunity to compensate.
>
> For the future improvement of reasoning path retrieval, we believe a feasible approach is to sample pseudo-topic entities from the KB based on the concept or other descriptions in the question, using them as the starting points for the walk. The intuition behind this is that entities belonging to the same concept may share similar relations. For example, in the question “What is the highest mountain in the world?”, we could sample “Mt. Fuji” and “Mt. Everest” as pseudo-entities based on the concept of "mountain," and then proceed with the search and filtering process to find the correct relation - “elevation above sea level.” This approach is plausible because our reasoning path involves only the relations.

---

> ### Author Response · Authors · 2024-11-29
> **Follow-up on Rebuttal Submission**
>
> Dear Reviewer,
>
> I hope you had an enjoyable and restful Thanksgiving. I’m writing to gently follow up on my rebuttal submission for the paper. I greatly appreciate the time and attention you’ve dedicated to reviewing my work, and I hope that my responses have sufficiently addressed your queries and concerns.
>
> If you’ve had the chance to revisit the rebuttal, I would be grateful if you could take into account the additional clarifications and improvements made. If possible, I would appreciate it if you could reconsider the score in light of the updated information.
>
> Thank you once again for your valuable time and consideration.

---

> > ### Comment · Reviewer_pGPC · 2024-12-03
> > **Reply**
> >
> > Thanks for your reply, but I think the comments in my review (mainly about writing) largely still stand and so I will keep the same score.

---

### Official Review · Reviewer_viu4 · 2024-11-04

**Soundness:** 3
**Presentation:** 3
**Contribution:** 2
**Rating:** 6
**Confidence:** 4

**Summary:**

This paper introduces REPANA, a framework for reasoning path navigated program induction that enables Large Language Models (LLMs) to reason over heterogeneous knowledge bases (KBs) with low-resource data. REPANA deals with the challenges of program induction. Usually, this requires a large amount of high-quality parallel data for a particular KB. It also addresses the poor transferability of models trained on single datasets because of the heterogeneity of question and KB schema. The experiments demonstrate that REPANA has strong generalization and transferability, outperforming other state-of-the-art low-resource methods and approaching the performance of supervised methods.

**Strengths:**

-	The proposed REPANA decouples the program generation capability into perceiving the KB and mapping questions to program sketches, which is an effective way.
-	REPANA exhibits strong transferability and generalization capabilities. This is an advantage for low-resource KBs where annotated data is scarce.

**Weaknesses:**

-	REPANA 's performance is affected by the lack of topic entities in the question, which can limit its effectiveness in scenarios where the question does not provide a clear starting point for reasoning.
-	REPANA's path navigation is prone to errors in questions with longer inference chains.
-	Although REPANA aims to alleviate the need for high-quality parallel data, it still relies on some amount of annotated data from rich-resource KBs.

**Questions:**

1.	How does REPANA handle questions that involve ambiguous or multiple root entities, and what is the impact on the accuracy of the retrieved reasoning paths?

---

> ### Author Response · Authors · 2024-11-21
> **Reply**
>
> We appreciate your valuable suggestions, and we reorganize your questions as follows:
>
> Q1: Regarding questions that do not have a topic entity.
>
> The inability to handle questions without topic entities is a common problem faced by many current approaches that involve searching on knowledge graphs. For example, as mentioned in the paper, StructGPT [1], Pangu [2], and ToG [3] view the process of walking on the graph as occurring simultaneously with reasoning. Therefore, when they cannot walk (due to the lack of a starting point), the reasoning naturally fails. However, our approach is two-stage, and the walk on the graph in the first stage primarily serves to augment the generation in the second stage. Therefore, theoretically, there are two advantages: (1) We can easily improve the first retrieval stage without changing the overall framework, as we have planned and mentioned in the conclusion of the paper. (2) The second stage has the opportunity to compensate for any failures in the first stage.
>
> For the future improvement of reasoning path retrieval, we believe a feasible approach is to sample pseudo-topic entities from the KB based on the concept or other descriptions in the question, using them as the starting points for the walk. The intuition behind this is that entities belonging to the same concept may share similar relations. For example, in the question “What is the highest mountain in the world?”, we could sample “Mt. Fuji” and “Mt. Everest” as pseudo-entities based on the concept of "mountain," and then proceed with the search and filtering process to find the correct relation - “elevation above sea level.” This approach is plausible because our reasoning path involves only the relations.
>
> Q2: Regarding questions with multiple roots
>
> Sorry we didn't state this more clearly in the paper, but we argue that our system has the potential to handle complex questions. By "complex," we assume you mean questions that have multiple topic entities, which result in multiple branches in the reasoning path (or tree). In fact, we did consider this situation, and the solution we currently adopt is to retrieve the reasoning path for each topic entity. For example, for the question, "Who is taller, James's son or O'Neal's son?", we retrieve two paths: [LeBron James, <start>, son, height] and [O'Neal, <start>, son, height]. In the current version of Repana, we simply pass these two paths as a list to the parser.
>
> We understand that the two paths could potentially be merged into a tree, perhaps in the form of ((LeBron James, <start>, son) (O'Neal, <start>, son) height). However, we would need to teach the LLMs to understand the meaning of this format, and there are too many possible ways to merge multiple paths. Therefore, we ultimately decided to leave this task to the second stage, where the model can be trained to understand the meaning of the paths we provide.
>
> Q3: require annotated data from rich-resource KBs
>
> Yes, our approach requires annotated data from rich-resource KBs. But considering that they are already there for use, we aim to avoid annotating more data on target low-resource KBs.
>
> [1] Structgpt: A general framework for large language model to reason over structured data.
>
> [2] Don't Generate, Discriminate: A Proposal for Grounding Language Models to Real-World Environments.
>
> [3] Think-on-graph: Deep and responsible reasoning of large language model with knowledge graph.

---

> ### Author Response · Authors · 2024-11-29
> **Follow-up on Rebuttal Submission**
>
> Dear Reviewer,
>
> I hope you had a relaxing Thanksgiving holiday. I wanted to follow up on my rebuttal submission for this paper. I truly appreciate the time and effort you’ve invested in reviewing my work, and I trust that my responses have addressed your questions and concerns comprehensively.
>
> If you’ve had the opportunity to review the rebuttal, I would be grateful if you could consider the additional clarifications and improvements made. If possible, I would appreciate it if you could re-evaluate your score based on the updates provided.
>
> Thank you once again for your time and thoughtful consideration.

---

### Official Review · Reviewer_NMDe · 2024-11-04

**Soundness:** 3
**Presentation:** 3
**Contribution:** 2
**Rating:** 6
**Confidence:** 4

**Summary:**

The paper introduces REPANA, a program induction framework designed to improve reasoning over heterogeneous knowledge bases (KBs) by leveraging a structured, two-part framework. REPANA separates the process of knowledge base schema perception from question-to-program mapping to improve Large Language Model (LLM) performance in low-resource KBs. It integrates two core components: a KB navigator, which retrieves reasoning paths, and a KB-agnostic parser that generates program sketches, allowing models to generalize across various KBs without extensive training on each. REPANA demonstrates superior generalization by performing well on unseen datasets, even approaching supervised methods in certain metrics.

**Strengths:**

- Decouples KB schema perception from program generation, allowing for greater transferability across different KBs.
- Performs effectively without large annotated datasets, alleviating the need for extensive manual annotations.
- Adaptable to varying KB structures, making it effective across diverse datasets with minimal retraining.

**Weaknesses:**

- Relies on topic entities for effective path retrieval, which limits performance in questions lacking clear topic entities.
- Accuracy declines with longer reasoning paths or multi-hop questions.
- It can hardly handle complex questions since the framework mainly focuses on exploring single reasoning chains.
- This method may be time costly since it utilizes LLMs multiple times in every search step.
- What is the relation of this work and other KBQA work which conducts pruning then answering (please list one such representative work as well if possible)?

**Questions:**

- How does REPANA handle questions without clear root entities effectively in various KBs?
- Why do you use the dev set to test your model in Table 2? Although you claim that questions during testing do not overlap with the training set, it is fairer to compare the performance of the same test set.
- Have you tried directly performing relation and entity filtering on ChatGPT instead of on tuned small Llama models?
- What is the performance of LLMs, like ChatGPT4 and Llama3-70b, on the datasets in Table 1 with few-shot demonstrations?
- The majority of MetaQA results in Table 1 are over 90, and half of the results are over 95. The dataset may not validate for differencing model performance.
- The format of Figure 3 looks weird.
- Appendix A should be removed.
- what is the cost of additional instruction tuning and what is the size of the dataset (did you mention this in the paper)?

---

> ### Author Response · Authors · 2024-11-21
> **Reply part1**
>
> Thank you for your valuable suggestions. We reorganize your questions as follows:
>
> Q1: Regarding questions that do not have a topic entity.
>
> The inability to handle questions without topic entities is a common problem faced by many current approaches that involve searching on knowledge graphs. For example, as mentioned in the paper, StructGPT [1], Pangu [2], and ToG [3] view the process of walking on the graph as occurring simultaneously with reasoning. Therefore, when they cannot walk (due to the lack of a starting point), the reasoning naturally fails. However, our approach is two-stage, and the walk on the graph in the first stage primarily serves to augment the generation in the second stage. Therefore, theoretically, there are two advantages: (1) We can easily improve the first retrieval stage without changing the overall framework, as we have planned and mentioned in the conclusion of the paper. (2) The second stage has the opportunity to compensate for any failures in the first stage.
>
> For the future improvement of reasoning path retrieval, we believe a feasible approach is to sample pseudo-topic entities from the KB based on the concept or other descriptions in the question, using them as the starting points for the walk. The intuition behind this is that entities belonging to the same concept may share similar relations. For example, in the question “What is the highest mountain in the world?”, we could sample “Mt. Fuji” and “Mt. Everest” as pseudo-entities based on the concept of "mountain," and then proceed with the search and filtering process to find the correct relation - “elevation above sea level.” This approach is plausible because our reasoning path involves only the relations.
>
> Q2: Regarding more complex questions
>
> Sorry we didn't state this more clearly in the paper, but we argue that our system has the potential to handle complex questions. By "complex," we assume you mean questions that have multiple topic entities, which result in multiple branches in the reasoning path (or tree). In fact, we did consider this situation, and the solution we currently adopt is to retrieve the reasoning path for each topic entity. For example, for the question, "Who is taller, James's son or O'Neal's son?", we retrieve two paths: [LeBron James, <start>, son, height] and [O'Neal, <start>, son, height]. In the current version of Repana, we simply pass these two paths as a list to the parser.
>
> We understand that the two paths could potentially be merged into a tree, perhaps in the form of ((LeBron James, <start>, son) (O'Neal, <start>, son) height). However, we would need to teach the LLMs to understand the meaning of this format, and there are too many possible ways to merge multiple paths. Therefore, we ultimately decided to leave this task to the second stage, where the model can be trained to understand the meaning of the paths we provide.
>
> Q3: Regarding the time cost.
>
> In fact, the time overhead is not a significant issue. Assuming we use random selection for entity filtering, as mentioned in the paper, there is only one API call per walking iteration. Including the final ranking operation, there are N+1 calls for an N-step walk.
>
> Additionally, to reduce the size of candidates for the input to the LLMs in each turn, we adopt some heuristic rules. For example, for relations in Freebase, we filter out those that start with ['type/object/name', 'common/topic/description', 'type/object/type', …], a list of relation domains that we consider unhelpful. As we have observed, the number of candidates after relation filtering typically ranges from 20 to 50, which is not a large list, and LLMs can process it quickly..
>
> Q4: Regarding relation between pruning then answering work
>
> We are not sure which kind of work you are referring to exactly. We assume that by "pruning," you are referring to works that adopt reranking or pruning during retrieval. One example might be RnG-KBQA [4], which generates all valid logical form candidates and uses a trained reranker to select those most relevant to the question. Compared to our work, their approach is also a two-stage process—retrieve and generate—but the first stage is too brute-force and does not provide the retriever with any reward, resulting in significant time and memory overhead. In contrast, our retriever is guided by LLMs and is much lighter.
>
> Another type of pruning is performed simultaneously with retrieval, as in Pangu[2]. Similar to our approach, they ask LLMs to filter and prune incorrect expansions. However, as we mentioned in Q1 and Q2, their reasoning process occurs simultaneously with the graph traversal, which is technically impossible to handle situations where questions have no topic entity or multiple topic entities. Additionally, their performance is relatively low under the ICL setting, especially when the number of demonstrations is fewer than 1000, which may not be available in low-resource KBs.

---

> ### Author Response · Authors · 2024-11-21
> **reply part2**
>
> Q5: Why dev set in table 2
>
> As we mentioned in the paper, we want to ensure that the schema items unseen during training are evaluated. Since we do not know the exact schema items that will be used in the test set (only contains the natural language questions and answers), we instead use the dev set. We then filter out examples that overlap with the training examples by using the annotated programs.
>
> Q6: Why not performing relation filtering on ChatGPT instead of on tuned Llama
>
> We believe this is a misunderstanding. We did use ChatGPT-3.5-turbo to perform the filtering (mentioned in Section 5.4 Implementation). The parser is tuned Llama.
>
> Q7: Regarding few-shot demonstration performance
>
> Sorry we didn’t consider the few-shot in-context learning approach. Some of the previous works [2][5][6] have explored this, demonstrating that the performance of current LLMs is not good enough in generating formal languages, unless provided with a large number of demonstrations, such as 1,000. We believe it is much more efficient to train a LoRA for Llama to learn how to generate logical forms.
>
> Q8: MetaQA not differencing model performance
>
> Yes, we agree with your point. We include this dataset because it is built on the movie domain, and we want to verify its transferability to specific domains. Additionally, it is a commonly used classic dataset. The results can at least demonstrate that our method is comparable to others.
>
> Q9: Regarding cost of additional instruction tuning and the size of the dataset
>
> We mentioned the size of our training set in Section 5.1. For evaluation, we use 500 randomly sampled examples from each dataset's test set.
>
> Regarding instruction tuning, there is no "additional" tuning. We perform instruction tuning to train the LoRA for Llama as the parser. GPU usage and parameter details are mentioned in Section 5.4. The total training time is approximately 5 hours per gpu.
>
> As for your other suggestions, such as those regarding Figure 3 and Appendix A, we realize that there were some mistakes made when we wrote in a hurry. We will do our best to address these issues in the next version.
>
> [1] Structgpt: A general framework for large language model to reason over structured data.
>
> [2] Don't Generate, Discriminate: A Proposal for Grounding Language Models to Real-World Environments.
>
> [3] Think-on-graph: Deep and responsible reasoning of large language model with knowledge graph.
>
> [4] Rng-kbqa: Generation augmented iterative ranking for knowledge base question answering.
>
> [5] Few-shot in-context learning for knowledge base question answering.
>
> [6] How Proficient Are Large Language Models in Formal Languages? An In-Depth Insight for Knowledge Base Question Answering.

---

> ### Author Response · Authors · 2024-11-29
> **Follow-up on Rebuttal Submission**
>
> Dear Reviewer,
>
> I hope you had a restful Thanksgiving holiday. I want to kindly follow up on my rebuttal submission for this paper. I appreciate the time and effort you have taken to review my work, and I hope my responses addressed your questions and concerns thoroughly.
>
> If you have had the chance to review the rebuttal, I would be grateful for your consideration of the additional clarifications and improvements we provided. If possible, I would appreciate it if you could re-evaluate your score in light of the clarifications.
>
> Thank you again for your time and consideration.

---

### Meta-Review · Area_Chair_FqAy · 2024-12-25

**Metareview:**

The paper introduces REPANA, a program induction framework designed to improve reasoning over heterogeneous knowledge bases (KBs) by leveraging a structured, two-part framework. REPANA separates the process of knowledge base schema perception from question-to-program mapping to improve Large Language Model (LLM) performance in low-resource KBs. It integrates two components: a KB navigator and a KB-agnostic parser that generates program sketches, allowing models to generalize across various KBs without extensive training on each. REPANA demonstrates superior generalization by performing well on unseen datasets, even approaching supervised methods in certain metrics.

Reviewers understood the paper and like the problem it tackles (NMDe, viu4, pGPC
) and its low-resource performance (NMDe, viu4, pGPC, 2g3F). Reviewers found the paper to be clear enough but the writing a rough (pGPC, 2g3F). However, no reviewers were excited about the results and gave non-committal reviews with ratings of 5 or 6. Reviewers did not speak to the importance of the problem and evaluate the significance of the improvements nor if the proposed method is innovative enough. It seems to fall slightly short for a top ML conference for a KB application paper.

**Additional Comments On Reviewer Discussion:**

authors responded to all reviewers and 2 reviewers acknowledged.

---

### Decision · Program_Chairs · 2025-01-22

Reject